# Large Unstained Cells (LUC): A Novel Predictor of CDK4/6 Inhibitor Outcomes in HR+ HER2-Negative Metastatic Breast Cancer

**DOI:** 10.3390/jcm14010173

**Published:** 2024-12-31

**Authors:** Furkan Ceylan, Mirmehdi Mehdiyev, Didem Şener Dede, Safa Can Efil, Ateş Kutay Tenekeci, Burak Bilgin, Şebnem Yücel, Hayriye Tatlı Doğan, Mehmet Ali Nahit Şendur, Muhammed Bülent Akıncı, Doğan Uncu, Bülent Yalçın

**Affiliations:** 1Department of Medical Oncology, Ankara Bilkent City Hospital, Ankara 06800, Turkey; 2Department of Medical Oncology, Ankara Yıldırım Beyazıt University, Ankara 06800, Turkey; 3Faculty of Medicine, Hacettepe University, 06230 Ankara, Turkey; 4Department of Pathology, Ankara Bilkent City Hospital, Ankara 06800, Turkey

**Keywords:** LUC, large unstained cells, breast cancer, prognosis, CDK 4/6 inhibitors

## Abstract

**Background**: Although CDK4/6 inhibitors combined with endocrine therapies have improved outcomes in HR+ HER2-negative metastatic breast cancer, predictive biomarkers for treatment response and adverse effects remain limited. This study assessed the prognostic and predictive value of large unstained cells (LUC), a subset of white blood cells that may reflect immune status or treatment response. **Methods**: A retrospective analysis of 210 patients with HR+ HER2-negative metastatic breast cancer treated with CDK 4/6 inhibitors between 2021 and 2024 was conducted. Clinical data, including demographics, tumor characteristics, and treatment regimens, were analyzed. Based on LUC levels, progression-free survival (PFS), overall survival (OS), and adverse events were evaluated. **Results**: The cohort had a median age of 57, of which 78% were postmenopausal. Common metastatic sites included bone (67%) and liver (24%). At a median follow-up of 18.5 months, the PFS and OS rates were 65% and 83%. Patients with low LUC levels had significantly shorter PFS (OR: 1.91; *p* = 0.014) and OS (OR: 2.39; *p* = 0.012), while high LUC levels correlated with a lower incidence of grade 3 neutropenia (OR: 0.49; *p* = 0.017). Liver metastasis and prior treatments were also linked to shorter survival. **Conclusions**: LUC levels emerge as a promising biomarker for predicting survival outcomes and the risk of neutropenia in HR+ HER2-negative metastatic breast cancer patients treated with CDK 4/6 inhibitors and endocrine therapy, showing their potential to guide personalized treatment approaches.

## 1. Introduction

Breast cancer is the most frequently diagnosed malignancy in women worldwide and ranks among the leading causes of cancer-related mortality [1]. According to GLOBOCAN 2022 data, 2.3 million women are diagnosed with breast cancer annually, and 665,000 women die from breast cancer-related causes [2]. Despite advancements in early detection and therapeutic interventions, breast cancer remains a significant public health concern. Improvements in screening methods and treatment strategies have contributed to a decline in breast cancer-related mortality rates [3]. The majority of breast cancer cases are diagnosed at an early stage. However, the presence of risk factors such as increased tumor size, lymph node involvement, or high tumor grade significantly elevates the likelihood of recurrence, with 20-year recurrence rates reaching up to 40% [4]. More than half of these recurrences manifest as distant metastases [5]. The incidence of de novo metastatic breast cancer is lower in developed countries, with approximately 5–10% of patients presenting with metastatic disease at diagnosis and a 5-year survival rate of around 25% [6].

Breast cancer exhibits substantial heterogeneity, characterized by diverse molecular and morphological subtypes that determine distinct biological behaviors, clinical presentations, and prognostic outcomes. Based on immunohistochemical evaluation of estrogen receptors (ERs), progesterone receptors (PRs), and HER2/Neu expression, breast cancer is classified into four subtypes. Approximately 70% of cases are hormone receptor-positive (HR+) and HER2-negative, a subtype effectively treated with endocrine therapy in both adjuvant and metastatic settings. Dysregulation of the cyclin D–CDK4/6–pRb axis has been identified as a key mechanism of resistance to endocrine therapy in HR+ HER2-negative breast cancer [7]. The addition of CDK4/6 inhibitors to endocrine therapy has significantly improved both PFS and OS in this patient population, establishing this combination as the standard of care for advanced HR+ HER2-negative breast cancer.

While advancements in systemic therapies have significantly improved outcomes for metastatic breast cancer, the contribution of locoregional interventions to survival remains limited [8]. The integration of CDK4/6 inhibitors with endocrine therapy has extended overall survival to nearly five years, offering a transformative approach for patients with HR+ HER2-negative disease [9]. Despite these strides, a critical gap persists in identifying reliable biomarkers to optimize treatment outcomes. Current efforts to predict therapeutic efficacy have explored factors such as HER2 status, the prognostic nutritional index (PNI), and sarcopenia status; however, these markers remain insufficient [10,11,12]. The need for robust and clinically actionable biomarkers to guide the effective use of CDK4/6 inhibitors and predict patient response remains unmet, highlighting an essential area for further research.

Hematologic toxicities represent the most significant adverse effects limiting the clinical utility of CDK4/6 inhibitors. The incidence and severity of these toxicities vary across agents due to differences in their inhibitory activity on CDK4 and CDK6. Palbociclib and ribociclib are associated with a higher frequency of hematologic toxicities compared to abemaciclib [13]. Given the prevalence and clinical implications of these adverse events, there is an urgent need to identify reliable biomarkers capable of predicting their occurrence. To date, no biomarker has been established to predict hematologic toxicities associated with CDK4/6 inhibitors.

LUC, identified in complete blood counts, represents an uncharted cell population that may serve as an indicator of the systemic inflammatory response. Detected using automated hemogram analyzers, LUC are defined as large, peroxidase-negative cells that remain unclassified within traditional hematologic frameworks. These cells encompass a heterogeneous population, potentially including peroxidase-negative lymphoblasts, activated lymphocytes, plasma cells, hairy cells, blasts, and hematopoietic stem cells [14,15]. Ongoing studies employing flow cytometry aim further to characterize the precise nature and functional role of LUC.

LUC has garnered increasing interest as a potential biomarker, with emerging evidence linking its levels to various biological and clinical contexts (Appendix A). Studies have demonstrated a correlation between LUC levels and CD8+ T lymphocytes expressing CD38 in patients with HIV, suggesting an association between LUC and adaptive immune responses [16]. Serum T lymphocytes and macrophages have been extensively studied as key components of the immune response to tumors; however, LUC, characterized by their peroxidase-negative nature and generally larger size, exhibit distinct physical properties that set them apart from these immune cells. These unique attributes underscore the potential of LUC as a promising biomarker of immune activity. Consequently, further investigation into this cell population is of significant importance to better understand its role in immune regulation and its potential clinical applications [17,18]. The clinical significance of LUC extends beyond their role in immune response, as studies have established a notable association between LUC levels and CD34+ hematopoietic stem cell counts in patients undergoing autologous stem cell transplantation [19]. This correlation highlights the potential of LUC as a predictive biomarker for hematologic toxicities, particularly in the context of therapies like CDK4/6 inhibitors, which exert their effects by halting cell proliferation and inducing widespread hematological side effects. This finding emphasizes the broader implications of LUC in anticipating treatment-related toxicities and optimizing therapeutic strategies. The significance of LUC is also evident in cancer, where elevated levels have demonstrated notable prognostic value in hematologic malignancies, including acute leukemia, chronic lymphocytic leukemia, and plasma cell leukemia, underscoring their potential in disease stratification and outcome prediction [20,21,22]. Beyond hematologic disorders, high LUC levels have also been identified in non-hematologic conditions, such as melanoma, viral infections, and systemic inflammatory diseases like vasculitis [23,24,25,26]. These findings collectively highlight the diverse and multifaceted role of LUC as a biomarker, reflecting complex inflammatory and hematopoietic processes. This dual capacity positions LUC as a critical tool with promising applications in cancer prognosis, immune monitoring, and the prediction of treatment-related toxicities.

With their ability to be measured using automated devices and highly standardized techniques, coupled with evidence from prior studies, LUC emerges as a candidate for biomarker development. In patients with metastatic cancer, LUC detected in peripheral blood may reflect the presence of inflammatory cells and hematopoietic stem cells. These cells likely signify the body’s inflammatory response or hematopoietic activity, potentially linking LUC levels to tumor progression or the body’s attempts to counteract malignancy. Given their role in the inflammatory response, LUC possess considerable prognostic potential. Furthermore, as a heterogeneous population containing various precursor cells, LUC also hold promise as predictors of hematologic toxicities associated with CDK4/6 inhibitors, adding further value to their clinical applicability.

In this study, we aim to explore the prognostic and predictive potential of large unstained cells in the context of metastatic HR+ HER2-negative breast cancer, focusing specifically on patients undergoing treatment with CDK4/6 inhibitors. By investigating the role of LUC, we aim to uncover novel insights that could enhance patient stratification, optimize therapeutic outcomes, and provide a deeper understanding of treatment-related hematologic toxicities in this patient population

## 2. Patients-Methods

### 2.1. Patient Selection

Patients with hormone receptor-positive (HR+) HER2-negative metastatic breast cancer who were treated with CDK 4/6 inhibitors and endocrine therapy at Ankara Bilkent City Hospital between January 2021 and October 2024 were retrospectively analyzed. HR testing (ER and PR) by IHC has been performed on metastatic breast cancer using the methodology outlined in the ASCO/CAP HR testing guideline. Breast cancers with 1–100% of cells positive for ER expression are considered ER-positive. This single-center, retrospective study utilized medical records to gather data on demographic and clinical parameters, including PFS and OS. The follow-up duration was calculated using the reverse Kaplan–Meier method. The study’s primary endpoints were PFS and OS in relation to LUC levels, while the secondary endpoint focused on the incidence of hematologic toxicity associated with varying LUC levels.

### 2.2. LUC Definition, Analysis and Sampling Timing

LUC is a distinct population of morphologically unclassified cells identified during automated complete blood count (CBC) analysis systems. These cells remain uncategorized by the analyzer due to their atypical characteristics and encompass a heterogeneous group that may include lymphoblasts, activated lymphocytes, or atypical lymphocytes. All blood samples were analyzed within 2 h of collection to prevent cellular degradation and maintain accuracy. Routine calibration and quality control procedures, as recommended by the manufacturer, were performed daily on the analyzer to ensure reliability. Analytical performance was periodically validated using control samples to monitor consistency. LUC levels were measured from peripheral blood at the time of diagnosis, specifically within three days prior to the initiation of systemic therapy.

### 2.3. Determination of LUC Levels

LUC levels were analyzed using an automated hematology analyzer (Siemens Medical Solutions Diagnostics, ADVIA 2120i, Tarrytown, NY, USA), a high-throughput system designed for the accurate and reproducible classification of white blood cells. The analyzer employs advanced optical and staining technologies to detect and categorize blood cell populations, with the following key steps applied to identify LUC:Peroxidase Method:

Blood samples undergo staining with a peroxidase reagent, enabling the classification of white blood cells (WBC) based on peroxidase activity. LUC are identified as peroxidase-negative cells, distinguishing them from granulocytes, monocytes, and other peroxidase-positive WBC.

Basophil/Lobularity Method:

Following the lysis of red blood cells (RBC), the analyzer assesses white blood cells using a two-angle laser scatter technique. This method evaluates cellular size and internal complexity, allowing the identification of LUC as large, unclassified cells that do not fit conventional leukocyte profiles.

### 2.4. Measurement and Reporting

LUC levels are expressed as a percentage of the total WBC count and absolute number, with all values directly derived from the analyzer’s software output. The ROC curve could not obtain a threshold value for LUC. In previous studies, no established cut-off value for LUC has been identified. To ensure consistency, patients in this study were stratified into two groups based on the median LUC value. Based on this threshold, patients were stratified into two groups: low LUC (<0.11) and high LUC (≥0.11). This classification facilitated the evaluation of the relationship between LUC levels and disease progression, as well as their prognostic significance for survival outcomes.

Low LUC Group: LUC levels < 0.11 K/μL.High LUC Group: LUC levels ≥ 0.11 K/μL.

### 2.5. Evaluation of Survival and Adverse Effects

This study aimed to evaluate the primary endpoints of PFS and OS in relation to LUC levels, using the median LUC value as the cutoff point. Progression-free survival is defined as the duration from the initiation of the CDK 4/6 inhibitor combined with endocrine therapy until disease progression or death, while overall survival is defined as the duration from the start of treatment to death or the last follow-up visit.

In addition, the secondary endpoint focused on analyzing the frequency of hematologic toxicity associated with different LUC levels. Toxicity was assessed based on the National Cancer Institute Common Terminology Criteria for Adverse Events (CTCAE), version 5.0. Blood samples were obtained at various intervals during treatment, and data on clinical examinations, hemograms, and biochemistry tests were systematically recorded. The toxicity analysis used the highest observed values from the patient records.

### 2.6. Statistics

Descriptive statistics were reported as mean and standard deviation for normally distributed numerical variables and as median and interquartile range for non-normally distributed variables. Categorical variables were presented as counts or frequencies. The Student’s *t*-test was utilized for comparisons of normally distributed numerical variables, while the Mann–Whitney U test was applied for non-normally distributed variables. Proportions in categorical variables were compared using the Chi-Square test. PFS and OS were analyzed using the Kaplan–Meier method, with intergroup comparisons performed via the log-rank test. Cox regression analysis was employed to identify independent predictors of survival, and variables with a *p*-value of less than 0.20 in the univariate analysis were included in the multivariate analysis. All statistical analyses were conducted using SPSS version 26.0 (SPSS Inc., Chicago, IL, USA), with a *p*-value of less than 0.05 considered statistically significant.

The institutional review board of Ankara Bilkent City Hospital granted ethical approval for this research, and the study was conducted in accordance with the principles outlined in the Declaration of Helsinki.

## 3. Results

### 3.1. Patients and Tumor Characteristics

A total of 210 patients were included in the study, with a median follow-up duration of 18.5 months. Patient and tumor characteristics are summarized in Table 1. The median age at diagnosis was 57 years, and 78% of the patients were postmenopausal, while 21% were premenopausal. Additionally, 90% of the patients had an ECOG performance status (PS) of 0/1. HER2 immunohistochemistry (IHC) results showed that 78% of the patients had a score of 0, 15% had a score of 1, and 7% had a score of 2. The most common sites of metastasis were bone (67%), liver (24%), lymph nodes (23%), and lung (13%). Sixty-four percent of the patients had de novo metastatic disease. Twelve percent of patients (*n* = 25) had received chemotherapy in the previous year. In terms of treatment for metastatic disease, 163 patients (78%) were treatment-naïve, while 47 patients (22%) had received at least one prior line of therapy. None of the patients had previously been treated with CDK4/6 inhibitors. Aromatase inhibitors (AI) were used in 71% of the patients, while fulvestrant was administered to 29%. Among the CDK4/6 inhibitors, ribociclib was given to 138 patients (66%), and palbociclib to 72 patients (34%). In terms of treatment response, 16 patients (8%) achieved a complete response, while 102 patients (49%) had a partial response.

The median LUC was 0.11 (IQR: 0.085–0.15). Patient and tumor characteristics, administered treatments, and treatment responses were similar between patients with high and low LUC levels. However, the frequency of lung metastasis was higher in the high LUC group (19% vs. 7%, *p* = 0.018). (Appendix A).

### 3.2. Progression-Free Survival and Overall Survival

The median PFS and OS were not reached during the study period. At the 18.5-month mark, the PFS rate was 65%, and the OS rate was 83%. When stratified by LUC levels measured prior to the initiation of the CDK4/6 inhibitor and endocrine therapy, patients with higher LUC levels demonstrated significantly longer progression-free survival (OR: 0.56, 95% CI: 0.34–0.93, *p* = 0.027) (Figure 1) and overall survival (OR: 0.41, 95% CI: 0.21–0.80, *p* = 0.010) (Figure 2). In the multivariate analysis for PFS, low LUC levels (OR: 1.91, 95% CI: 1.14–3.20, *p* = 0.014), the presence of liver metastasis (OR: 3.18, 95% CI: 1.76–5.74, *p* < 0.001), and prior endocrine therapy (OR: 2.75, 95% CI: 1.49–5.05, *p* = 0.001) were associated with shorter PFS (Table 2). In the multivariate analysis for OS, low LUC levels (OR: 2.39, 95% CI: 1.21–4.71, *p* = 0.012) and the presence of liver metastasis (OR: 2.41, 95% CI: 1.23–4.75, *p* = 0.011) were associated with shorter OS (Table 3).

Kaplan–Meier curve showing progression-free survival (PFS) stratified by LUC levels in patients with metastatic HR+ HER2-negative breast cancer. At 18.5 months, the PFS rate was 65%. Patients with elevated LUC levels, measured prior to the initiation of the CDK4/6 inhibitor and endocrine therapy, had significantly longer PFS compared to those with lower LUC levels (OR: 0.56, 95% CI: 0.34–0.93, *p* = 0.027). LUC: large unstained cells; PFS: progression-free survival.

Kaplan–Meier curve illustrating overall survival (OS) stratified by LUC levels in patients with metastatic HR+ HER2-negative breast cancer. At 18.5 months, the OS rate was 83%. Patients with elevated LUC levels, measured prior to the initiation of the CDK4/6 inhibitor and endocrine therapy, demonstrated significantly longer OS compared to those with lower LUC levels (OR: 0.41, 95% CI: 0.21–0.80, *p* = 0.010). LUC: large unstained cells; OS: overall survival.

### 3.3. Adverse Effects

Hematologic toxicity occurred in 193 patients (92%), with grade ≥ 3 hematologic toxicity observed in 103 patients (49%). Neutropenia was present in 187 patients (89%), and grade ≥ 3 neutropenia was noted in 96 patients (46%). Elevated ALT levels were found in 48 patients (23%), with grade ≥ 3 ALT elevation occurring in only 4 patients (2%). Additionally, acute kidney injury was documented in 6 patients (3%), skin toxicity in 4 patients (2%), and QT prolongation in 3 patients (2%).

Patients with higher LUC levels experienced significantly lower rates of hematologic toxicity (87% vs. 98%, *p* = 0.008) and neutropenia (82% vs. 98%, *p* < 0.001). Similarly, grade ≥ 3 hematologic toxicity (41% vs. 61%, *p* = 0.006) and grade ≥ 3 neutropenia (39% vs. 56%, *p* = 0.015) were less common in patients with elevated LUC levels. The incidence of other adverse events was comparable across both groups (Table 4).

In the multivariate analysis for grade ≥ 3 neutropenia, ribociclib use (OR: 0.46, 95% CI: 0.25–0.84, *p* = 0.011), aromatase inhibitor therapy (OR: 0.51, 95% CI: 0.27–0.96, *p* = 0.048), and higher LUC levels (OR: 0.49, 95% CI: 0.27–0.88, *p* = 0.017) were independently associated with a lower risk of developing grade ≥ 3 neutropenia (Table 5).

## 4. Discussion

In this study, we demonstrated that elevated LUC levels are significantly associated with longer PFS and OS, as well as a reduced incidence of neutropenia and grade ≥ 3 neutropenia. Furthermore, our findings indicate that the presence of liver metastases is linked to shorter PFS and OS, while prior treatment for metastatic disease is associated with shorter PFS.

While locoregional therapies, such as surgery and radiotherapy, play a role in symptom control and local disease management, their contribution to overall survival in metastatic breast cancer remains limited [8]. The combination of CDK4/6 inhibitors and endocrine therapy represents the standard of care for the first-line treatment of metastatic HR+ HER2-negative breast cancer and remains effective in subsequent lines for patients who have not previously received this regimen. While clinical trials and real-world evidence demonstrate significant efficacy, disease progression remains an inevitable challenge. The search for reliable biomarkers to predict clinical outcomes, including survival and treatment-related adverse effects, continues to be a critical focus of oncology research. In this study, we demonstrated that LUC holds potential as a novel biomarker, offering valuable insights into its ability to predict clinical outcomes and hematologic toxicities in patients treated with CDK4/6 inhibitors. This finding highlights the importance of further exploration into LUC’s role in personalizing treatment strategies and improving outcomes for metastatic HR+ HER2-negative breast cancer.

In metastatic HR+ HER2-negative breast cancer, previous studies have identified several prognostic indicators, including higher HER2 immunohistochemistry scores, estrogen receptor (ER) expression levels, Ki67 index, poor differentiation, and the presence of an objective response [27,28,29,30,31,32]. Emerging biomarkers such as circulating tumor DNA (ctDNA) and circulating tumor cells (CTCs) have also gained substantial attention due to their ability to provide valuable insights into tumor dynamics. These markers show promise in predicting treatment efficacy, monitoring therapeutic response, and detecting resistance mechanisms in metastatic breast cancer [33,34]. While ctDNA and CTCs are highly informative, they require expensive and time-intensive processing. In contrast, LUC represents a simple, rapid, and cost-effective alternative. In our study, LUC was identified as a significant prognostic marker for both PFS and OS, addressing the critical need for accessible and affordable prognostic and predictive tools in HR+ HER2-negative breast cancer.

CDK4/6 inhibitors have been shown to modulate immune responses by inducing T cell activation while simultaneously suppressing T cell proliferation. This dual effect suggests a complex dynamic where CDK4/6 inhibitors may enhance the immune recognition of tumor cells but limit the expansion of T cell populations, potentially influencing the overall effectiveness of immune responses during treatment [35]. Moreover, findings from the PALOMA-2 study revealed that PD-L1 expression is an insufficient biomarker for detecting immune activation in HR+ HER2-negative breast cancer, showing the urgent need for more reliable indicators to assess immune dynamics in this specific subtype [36]. In this context, LUC emerges as a promising candidate, potentially representing precursor cells of cytotoxic T lymphocytes. Our study demonstrated a significant association between higher LUC levels and longer survival, suggesting that LUC may reflect a population of CD8+ T cells involved in immune activation. This observation supports the potential of LUC as a biomarker for predicting favorable clinical outcomes in HR+ HER2-negative breast cancer and highlights its relevance in the broader landscape of immunomodulatory biomarkers.

The administration of CDK4/6 inhibitors necessitates vigilant monitoring due to their well-documented hematological toxicities. Although no head-to-head randomized controlled trials have directly compared hematologic toxicity rates among different CDK4/6 inhibitors, pharmacovigilance studies indicate that palbociclib and ribociclib are associated with higher incidences of hematologic adverse events compared to abemaciclib [13]. Despite this, no reliable biomarker has yet been identified to predict the frequency or severity of these side effects.

In our study, we observed a significantly lower incidence of grade ≥ 3 neutropenia in patients with higher LUC levels. Given that LUC is thought to include hematopoietic stem cells, elevated LUC levels may reflect an enhanced hematopoietic capacity, thereby reducing susceptibility to neutropenia. This finding suggests that LUC could serve as a practical and accessible biomarker for predicting the risk of severe neutropenia, offering a valuable tool for mitigating one of the most common and impactful side effects of CDK4/6 inhibitor therapy.

The retrospective nature of our study introduces inherent limitations, including the potential for recall bias, selection bias, and the influence of confounding factors. Although our primary aim was to evaluate the prognostic and predictive role of LUC, the absence of flow cytometric analysis for the detailed characterization of LUC represents a notable limitation. Flow cytometry could have allowed for a more precise identification and functional understanding of these cells.

Despite these constraints, our study pioneers in demonstrating the prognostic significance of LUC in breast cancer. Furthermore, it stands out due to its inclusion of detailed subgroup analyses and its focus on identifying biomarkers that predict the efficacy and treatment responsiveness to CDK4/6 inhibitors. These findings provide a foundation for future studies to explore LUC as a novel, accessible biomarker in this context.

## 5. Conclusions

This study highlights the potential of LUC as a novel and accessible biomarker in metastatic HR+ HER2-negative breast cancer. By demonstrating the prognostic significance of LUC in predicting progression-free survival, overall survival, and the incidence of hematologic toxicities, our findings pave the way for its integration into routine clinical practice. The unique properties of LUC, including its ease of measurement using automated hematology analyzers and its potential association with immune and hematopoietic activity, position it as a promising tool for refining patient stratification and optimizing therapeutic strategies.

While the retrospective design and the absence of flow cytometric analysis represent limitations, this study is the first to establish a significant relationship between LUC levels and clinical outcomes in breast cancer. Future prospective studies are needed to validate these findings, further characterize LUC, and explore its potential in guiding personalized treatment approaches. LUC offers a cost-effective and readily available biomarker that holds significant promise in addressing unmet needs in the management of metastatic HR+ HER2-negative breast cancer.

## Figures and Tables

**Figure 1 jcm-14-00173-f001:**
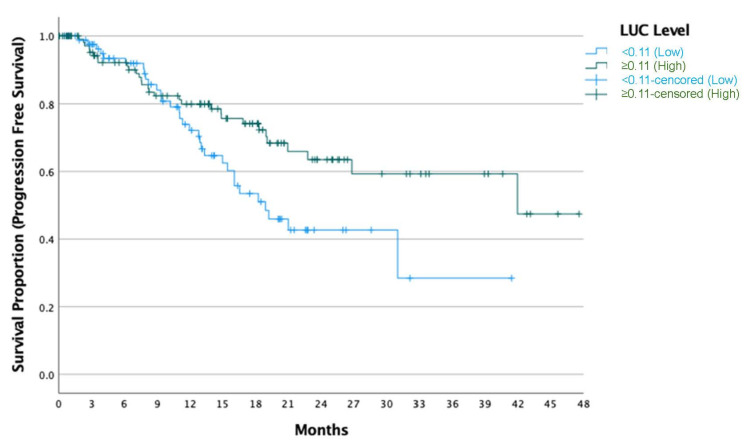
Progression-Free Survival (PFS) Stratified by LUC Levels.

**Figure 2 jcm-14-00173-f002:**
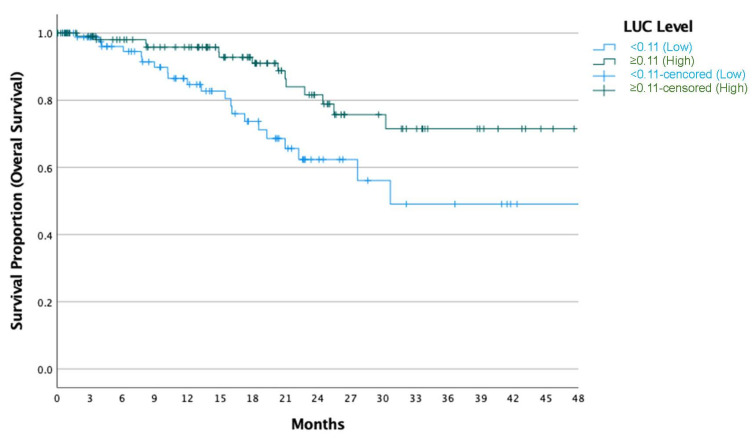
Overall Survival (OS) Stratified by LUC Levels.

**Table 1 jcm-14-00173-t001:** Baseline Clinical and Tumor Characteristics of the Patient Cohort.

Age, Median, IQR		57 (46–68)
Gender (Female)		207 (99%)
Menopause	Pre	43 (21%)
	Post	164 (78%)
	Male	3 (1%)
ECOG	0	10 (5%)
	1	177 (85%)
	2	17 (8%)
	3	6 (2%)
HER2 Status	0	158 (78%)
	1	31 (15%)
	2	14 (7%)
Metastatic Area	Bone	140 (67%)
	Lung	28 (13%)
	Liver	50 (24%)
	Lymph Node	48 (23%)
Received Chemotherapy in The Previous Year		25 (12%)
De novo Metastatic		134 (64%)
Line	First	163 (78%)
	Second and later	47 (22%)
Received LHRH		28 (13%)
Co-Administration ET	AI	149 (71%)
	Fulvestrant	61 (29%)
CDK 4/6	Ribociclib	138 (66%)
	Palbociclib	72 (34%)
Best Radiographic response	CR	16 (8%)
	PR	102 (49%)
	SD	50 (24%)
	PD	18 (9%)
	N/E	24 (10%)
Follow-up Time		18.5 months (16.6–20.4)
Progression		63 (30%)
Death		35 (17%)
PFS (Progression Free Survival)		N/E
OS (Overall Survival)		N/E

Baseline clinical and tumor characteristics of the patient cohort, including demographic details, menopausal status, ECOG performance score, HER2 status, metastatic sites, treatment history, and best radiographic response. Data are represented as counts and percentages, with age and follow-up time presented as median values with interquartile ranges (IQR). ECOG: Eastern Cooperative Oncology Group; HER2: human epidermal growth factor receptor 2; LHRH: luteinizing hormone-releasing hormone; ET: endocrine therapy; AI: aromatase inhibitor; CR: complete response; PR: partial response; SD: stable disease; PD: progressive disease; N/E: not evaluated.

**Table 2 jcm-14-00173-t002:** Univariate and Multivariate Analysis Results on Progression-Free Survival.

	Univariate Analysis			Multivariate Analysis		
	OR	95% CI	*p*	OR	95% CI	*p*
LUC < 0.11 vs. ≥0.11	1.77	1.07–2.93	0.027	1.91	1.14–3.20	0.014
Received Chemotherapy in the previous Year	1.70	1.00–2.91	0.052			
ECOG 0/1 vs. 3/4	0.70	0.22–2.23	0.541			
Ribociclib vs. palbociclib	0.56	0.54–0.93	0.023			
Menopause Post- vs. Pre-	0.94	0.50–1.77	0.851			
Age < 65 vs. ≥65	0.62	0.34–1.13	0.119			
Presence of Liver Metastasis	2.40	1.43–4.02	<0.001	3.18	1.76–5.74	<0.001
De novo Metastatic	0.65	0.39–1.08	0.096			
First Line vs. Second and Later	0.60	0.35–1.02	0.059	0.36	0.20–0.67	0.001

Univariate and multivariate analysis results for progression-free survival (PFS) in patients with metastatic HR+ HER2-negative breast cancer. The table presents odds ratios (OR), 95% confidence intervals (CI), and *p*-values for factors such as LUC levels, prior chemotherapy, ECOG performance status, type of CDK4/6 inhibitor (ribociclib vs. palbociclib), menopausal status, age, presence of liver metastasis, de novo metastatic status, treatment line (first vs. second and later), dose reduction, and treatment withdrawal. Significant predictors of shorter PFS in multivariate analysis include low LUC levels, liver metastasis, and later-line therapy. LUC: large unstained cells; ECOG: Eastern Cooperative Oncology Group; OR: odds ratio; CI: confidence interval.

**Table 3 jcm-14-00173-t003:** Univariate and Multivariate Analysis Results on Overall Survival.

	Univariate Analysis			Multivariate Analysis		
	OR	95% CI	*p*	OR	95% CI	*p*
LUC < 0.11 vs. ≥0.11	2.45	1.25–4.83	0.010	2.39	1.21–4.71	0.012
Received Chemotherapy in the previous year	1.28	0.50–3.32	0.607			
ECOG 0/1 vs. 3/4	1.07	0.26–4.50	0.924			
Ribociclib vs. palbociclib	0.57	0.29–1.13	0.105			
Menopause Post- vs. Pre-	1.79	0.63–5.07	0.277			
Age < 65 vs. ≥65	1.08	0.53–2.20	0.842			
Presence of Liver Metastasis	2.44	1.24–4.79	0.010	2.41	1.23–4.75	0.011
De novo Metastatic	1.80	0.85–3.85	0.128			
First Line vs. Second and Later	0.86	0.39–1.89	0.699			

Univariate and multivariate analysis results for overall survival (OS) in patients with metastatic HR+ HER2-negative breast cancer. The table presents odds ratios (OR), 95% confidence intervals (CI), and *p*-values for factors such as LUC levels, prior chemotherapy, ECOG performance status, CDK4/6 inhibitor type (ribociclib vs. palbociclib), menopausal status, age, liver metastasis, de novo metastatic status, treatment line, dose reduction, and treatment withdrawal. Significant predictors of shorter OS in multivariate analysis include low LUC levels, liver metastasis, dose reduction, and treatment withdrawal. Abbreviations: LUC: large unstained cells; ECOG: Eastern Cooperative Oncology Group; OR: odds ratio; CI: confidence interval.

**Table 4 jcm-14-00173-t004:** Adverse Effect According to The Level of LUC.

	LUC < 0.11 (*n* = 86)	LUC ≥ 0.11 (*n* = 119)	*p* Value	Whole Group
Hematologic toxicity (Whole Grade)	84 (98%)	104 (87%)	0.008	193 (92%)
Anemia (Whole Grade)	57 (66%)	76 (64%)	0.721	136 (65%)
Neutropenia (Whole Grade)	84 (98%)	98 (82%)	<0.001	187 (89%)
Thrombocytopenia (Whole Grade)	29 (34%)	29 (24%)	0.142	59 (28%)
Grade ≥ 3 Hematologic toxicity	52 (61%)	49 (41%)	0.006	103 (49%)
Grade ≥ 3 Anemia	8 (9%)	12 (10%)	0.852	20 (10%)
Grade ≥ 3 Neutropenia	48 (56%)	46 (39%)	0.015	96 (46%)
Grade ≥ 3 Thrombocytopenia	6 (7%)	7 (6%)	0.751	13 (6%)
ALT (Whole Grade)	19 (22%)	29 (24%)	0.704	48 (23%)
Grade ≥ 3 ALT increase	1 (1%)	3 (3%)	0.641	4 (2%)
Acute Kidney Injury	4 (5%)	2 (2%)	0.240	6 (3%)
QT prolongation	2 (2%)	1 (1%)	0.573	3 (2%)
Skin Toxicity	1 (1%)	3 (3%)	0.641	4 (2%)

Adverse effects were stratified by LUC levels, comparing patients with LUC < 0.11 and LUC ≥ 0.11. The table reports the incidence of hematologic toxicities, anemia, neutropenia, thrombocytopenia, ALT elevation, acute kidney injury, QT prolongation, and skin toxicity. Toxicities are further categorized into overall and grade ≥ 3 events. ALT: alanine aminotransferase; LUC: large unstained cells; QT: QT interval on an electrocardiogram.

**Table 5 jcm-14-00173-t005:** Univariate and Multivariate Analysis for grade ≥ 3 Neutropenia.

	Univariate Analysis			Multivariate Analysis		
	OR	95% CI	*p*	OR	95% CI	*p*
Age (>65 vs. <64)	1.47	0.81–1.67	0.210			
ECOG 0/1 vs. 2/3	0.87	0.35–2.17	0.768			
Line	0.54	0.28–1.05	0.069			
Received Chemotherapy In The Previous Year	0.92	0.40–2.13	0.839			
Ribociclib vs. Palbociclib	0.42	0.24–0.75	0.004	0.46	0.25–0.84	0.011
AI (AI vs. Fulvestrant)	0.52	0.28–0.94	0.031	0.51	0.27–0.96	0.038
Pre-menopause vs. post-menopause	0.62	0.31–1.24	0.178			
LUC < 0.11 vs. LUC ≥ 0.11	2.00	1.14–3.52	0.016	2.04	1.14–3.67	0.017

Univariate and multivariate analysis results for predicting the risk of grade ≥ 3 neutropenia in patients with metastatic HR+ HER2-negative breast cancer. The table reports odds ratios (OR), 95% confidence intervals (CI), and *p*-values for factors such as age, ECOG performance status, treatment line, prior chemotherapy, CDK4/6 inhibitor type (ribociclib vs. palbociclib), endocrine therapy (aromatase inhibitor vs. fulvestrant), menopausal status, and LUC levels. Significant predictors of lower risk of grade ≥ 3 neutropenia in the multivariate analysis include ribociclib use, aromatase inhibitors, and higher LUC levels. Abbreviations: LUC: large unstained cells; ECOG: Eastern Cooperative Oncology Group; OR: odds ratio; CI: confidence interval.

## Data Availability

The data supporting this study’s findings are available from the corresponding author upon reasonable request. In adherence to ethical standards and institutional policies, only fully anonymized datasets will be shared to protect patient confidentiality. Data access will be granted exclusively for legitimate academic and non-commercial research purposes, following a formal request and subject to approval in line with institutional and ethical regulations.

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
