# Peer review of "Large Unstained Cells (LUC): A Novel Predictor of CDK4/6 Inhibitor Outcomes in HR+ HER2-Negative Metastatic Breast Cancer"

_jcm, 2024, doi:10.3390/jcm14010173_

Round 1

Reviewer 1 Report

Comments and Suggestions for Authors

Large Unstained Cells (LUC): A Novel Predictor of CDK4/6 Inhibitor Outcomes in HR+ HER2-Negative Metastatic Breast Cancer

Overall Assessment:

This article is showcasing high standards in scientific writing and research. The clear presentation of methods and results, combined with a well-framed introduction and thoughtful discussion, makes this study a valuable resource for the consideration of LUC as predictive marker of CDK4/6 inhibitors in metastatic breast cancer.

Acceptance with minor revision is recommended.

General comments:

I would like to begin by congratulating the authors on their submission and commend them for their work. This article presents a well-structured clinical study, focusing on the predictive value of large unstained cells (LUC) regarding CDK4/6 inhibitor treatment in metastatic breast cancer.

The authors have done an excellent job of introducing the background and contextualizing the aim of the study. The background provided is concise but relevant, offering readers a clear understanding with logical introduction of breast cancer, CDK4/6 inhibitors, their roles in hormone receptor positive metastatic breast cancer and their adverse effects and lastly focusing on the large unstained cells as a potential predictive marker.

The authors have meticulously detailed their design and methods, ensuring that the study is well documented. Each step of the process is described with clarity.

The presentation of results is equally commendable. The authors have provided a clear and comprehensive account of their findings, supported by well-organized tables and figures. The narrative accompanying the data effectively guides the reader through the results, highlighting significant outcomes without overwhelming them with unnecessary details.

The discussion and conclusion sections are well-crafted, connecting the results back to the original research question and the broader implications. The authors successfully communicate the potential impact of their findings while acknowledging the limitations of their study.

Minor revisions:

After careful review of this manuscript see below the minor comments recommended before final acceptance:

Comment 1: In the introduction line 61-63, as the reference quoted does not imply the association of high LUC levels directly with immune response linked to cancer, could the authors rephrase the sentence to introduce this reference only associated with viral infection.

Comment 2: In order to enhance the reader flow and ease of understanding, I would recommend reordering the table numbering according to their use in the manuscript to follow a logical sequence. As it is currently appearing in the manuscript, the order is 1, 3, 4, 2, 5.

Comment 3: The authors mentioned in the methods section that the two group, low and high LUC were defined by the median value (Section 2.2, line 90-91), stipulating that low LUC group is defined by a value less or equal to the median (0.11). However, in the header of Table 2 (Page 7, below line 214) the low LUC group is defined as only less than 0.10. This comment is twofold, first could the authors correct the median value displayed in Table 2 low LUC to 0.11, secondly could the authors clarify the methods to be in accordance with Table 2 that suggest patients with a value of 0.11 to be included in the High LUC group.

Comment 4: Same comment as above for Figure 1 and 2, <0.10 should be replace with <0.11, moreover the clarity of the figures would be enhanced if the group names are spell out in the legend (i.e “low LUC (<0.11)”) in order to associate the colour to the name of the group.

Comment 5: In the results authors observed that a higher frequency of lung metastases was detected in the high LUC group, could the authors comment this observation in light of the better PFS and OS observed in this group compared to the low LUC group and how the two observations could be reconciled.

Comment 6: Font size across Tables is appearing to be different and should be harmonised to fit the font used in the first Table.

Comment 7: Line 44 delete space before reference 4

Comment 8: Line 57 reference 9 and 10 should be added at the end of the sentence with reference 8

Author Response

We would like to sincerely thank the reviewer for their thoughtful and encouraging feedback on our manuscript. We greatly appreciate the positive assessment of our work and the constructive suggestions provided. The reviewer’s insights have been extremely helpful in further improving the clarity, precision, and overall quality of the manuscript. We have carefully addressed each comment and incorporated the necessary revisions to ensure the manuscript meets the highest standards. We are confident that these changes have strengthened the work and made it clearer and more accessible to readers. Below, we provide a detailed point-by-point response to each comment.

Large Unstained Cells (LUC): A Novel Predictor of CDK4/6 Inhibitor Outcomes in HR+ HER2-Negative Metastatic Breast Cancer

Overall Assessment:

This article is showcasing high standards in scientific writing and research. The clear presentation of methods and results, combined with a well-framed introduction and thoughtful discussion, makes this study a valuable resource for the consideration of LUC as predictive marker of CDK4/6 inhibitors in metastatic breast cancer.

Acceptance with minor revision is recommended.

General comments:

I would like to begin by congratulating the authors on their submission and commend them for their work. This article presents a well-structured clinical study, focusing on the predictive value of large unstained cells (LUC) regarding CDK4/6 inhibitor treatment in metastatic breast cancer.

The authors have done an excellent job of introducing the background and contextualizing the aim of the study. The background provided is concise but relevant, offering readers a clear understanding with logical introduction of breast cancer, CDK4/6 inhibitors, their roles in hormone receptor positive metastatic breast cancer and their adverse effects and lastly focusing on the large unstained cells as a potential predictive marker.

The authors have meticulously detailed their design and methods, ensuring that the study is well documented. Each step of the process is described with clarity.

The presentation of results is equally commendable. The authors have provided a clear and comprehensive account of their findings, supported by well-organized tables and figures. The narrative accompanying the data effectively guides the reader through the results, highlighting significant outcomes without overwhelming them with unnecessary details.

The discussion and conclusion sections are well-crafted, connecting the results back to the original research question and the broader implications. The authors successfully communicate the potential impact of their findings while acknowledging the limitations of their study.

Minor revisions:

After careful review of this manuscript see below the minor comments recommended before final acceptance:

Comment 1: In the introduction line 61-63, as the reference quoted does not imply the association of high LUC levels directly with immune response linked to cancer, could the authors rephrase the sentence to introduce this reference only associated with viral infection.

Authors’ Response: We thank the reviewer for this valuable observation. We have carefully revised the relevant section of the Introduction to ensure that the reference is now accurately presented in the context of viral infections. This adjustment removes any unintended association with immune responses linked to cancer.

We believe this clarification has improved the accuracy and clarity of the manuscript.

Comment 2: In order to enhance the reader flow and ease of understanding, I would recommend reordering the table numbering according to their use in the manuscript to follow a logical sequence. As it is currently appearing in the manuscript, the order is 1, 3, 4, 2, 5.

Authors’ Response: We thank the reviewer for this suggestion aimed at improving the flow and readability of the manuscript. As recommended, we have carefully reordered the tables to ensure they are presented in a logical sequence that aligns with their first mention in the text.

We believe this adjustment enhances the clarity and ease of understanding for readers and improves the overall presentation of the manuscript.

Comment 3: The authors mentioned in the methods section that the two group, low and high LUC were defined by the median value (Section 2.2, line 90-91), stipulating that low LUC group is defined by a value less or equal to the median (0.11). However, in the header of Table 2 (Page 7, below line 214) the low LUC group is defined as only less than 0.10. This comment is twofold, first could the authors correct the median value displayed in Table 2 low LUC to 0.11, secondly could the authors clarify the methods to be in accordance with Table 2 that suggest patients with a value of 0.11 to be included in the High LUC group.

Authors’ Response: We thank the reviewer for identifying this discrepancy and bringing it to our attention. We have corrected the header in Table 2 to reflect the appropriate median value cutoff of 0.11. All references to LUC grouping throughout the manuscript, including the methods section, now clearly define the low LUC group as <0.11 and the high LUC group as ≥0.11.

This correction ensures consistency across the manuscript and aligns the methods with the results presented. We appreciate the reviewer’s careful review, which has improved the clarity and accuracy of our work.

Comment 4: Same comment as above for Figure 1 and 2, <0.10 should be replace with <0.11, moreover the clarity of the figures would be enhanced if the group names are spell out in the legend (i.e “low LUC (<0.11)”) in order to associate the colour to the name of the group.

Authors’ Response: We thank the reviewer for this helpful suggestion. As requested, we have corrected the cutoff value in Figure 1 and Figure 2 to reflect <0.11. Additionally, we have updated the figure legends to explicitly spell out the group names (e.g., “low LUC (<0.11)” and “high LUC (≥0.11)”), ensuring that the group labels are clearly associated with their respective colors.

We believe these adjustments significantly improve the clarity and visual interpretation of the figures, and we are grateful for the reviewer’s attention to detail.

Comment 5: In the results authors observed that a higher frequency of lung metastases was detected in the high LUC group, could the authors comment this observation in light of the better PFS and OS observed in this group compared to the low LUC group and how the two observations could be reconciled.

  • Authors’ Response: We thank the reviewer for this insightful comment, which allows us to address this important observation. While the presence of visceral metastases, including lung metastases, is typically associated with poorer survival outcomes, the prognosis can vary depending on the extent and distribution of metastases. Specifically, patients with isolated lung metastases tend to have better outcomes than those with metastases in multiple visceral organs. (PMID: 34906122)

In our cohort, the high LUC group exhibited a higher frequency of lung metastases; however, most of these patients had limited metastatic burden, including isolated lung involvement, which may have contributed to their better PFS and overall survival OS. Additionally, the presence of high LUC levels may itself reflect an enhanced immune or hematopoietic response, which could improve treatment outcomes even in patients with lung metastases, traditionally considered a poorer prognostic feature.

Importantly, these findings further shows the potential prognostic significance of LUC levels. Elevated LUC levels could serve as a biomarker of favorable immune or hematopoietic activity, potentially compensating for the adverse impact of visceral metastases in certain patients. This hypothesis aligns with the observed longer survival outcomes in the high LUC group and supports the role of LUC as a meaningful prognostic indicator.

Comment 6: Font size across Tables is appearing to be different and should be harmonised to fit the font used in the first Table.

Authors’ Response: We thank the reviewer for their keen attention to detail and for bringing this formatting issue to our attention. We have carefully reviewed all tables and harmonized the font size and style to match the formatting of the first table. This ensures consistency and improves the overall presentation quality of the manuscript.

Comment 7: Line 44 delete space before reference 4

Authors’ Response: We thank the reviewer for pointing out this formatting issue. We have removed the unnecessary space before reference 4 to ensure consistency and accuracy throughout the manuscript.

Comment 8: Line 57 reference 9 and 10 should be added at the end of the sentence with reference 8

Authors’ Response: We thank the reviewer for this observation. The Introduction has been entirely revised to improve its clarity and flow, and during this process, we carefully addressed the placement and relevance of references, including the ones mentioned. The revised section now appropriately integrates references to ensure accuracy and logical progression.

We are deeply grateful to Reviewer 1 for the kind words and encouraging assessment of our work. Your detailed suggestions on formatting, clarity, and interpretation have been instrumental in refining the presentation of our results. We hope that the changes we have made align with your expectations and further improve the manuscript.

Reviewer 2 Report

Comments and Suggestions for Authors

This study addresses an important clinical challenge: the identification of predictive and prognostic biomarkers in metastatic HR+ HER2-negative breast cancer. The investigation of large unstained cells (LUC) as a novel biomarker is an innovative concept, however the lack of information on origination of LUC, validation, characterization effects the robustness of the study. LUC is the center of focus of the study, and there are no example images of appearance of LUC and how are different from other blood cells.

Comments

1. The novelty of this study is that it explores previously underexplored biomarker LUC, however, there are lack of biological insights about the same. For example, the introduction lacks information on how LUCs are identified. Is it based on their size? LUC needs to be defined better in introduction.

2. Additionally, document all the LUC related studies and respective cancers in a tabular form, add all the outcomes seen in patients in literature in correlation to LUC. Introduction needs significant improvement.

3.In literature, has LUC been used for any prognostic value? Please re-write your introduction.

4. In the introduction, when you say HR+, do you mean both ER+ and PR+? Please clarify. Why was HR+ HER2- group selected specifically?

5. Patient selection: What stage were these patients? Did they receive neoadjuvant or adjuvant treatment? Did they undergo surgery? Please make this section more detailed. It lacks a lot of important information.

6. Definition of LUC and sampling time: Same comments as introduction.

-  Explain in detail how are LUCs sampled. How do you get the cutoff 0.11 number? How is it meaningful?

- Are LUCs CAMLs?

-Are there any specific biomarkers/antibodies that are used to identify LUCs?

- What is the biological insight/function of LUCs in cancer or treatment response?

-What is the size range of LUCs? Is there a possibility of any false negatives or false positives?

- What are the different techniques for identifying LUCs? How accurate are they? Which technique was used here and why?

- How do you ensure what you are counting is LUC?

- Is it reproducible in other lab settings to identify LUC? Please give details on your protocols and methodology to help understand better how LUCs are defined.

- No images of LUCs added in paper. Add several images of LUC, validate with biomarker testing if this study is looking at LUCs indeed. How do you differentiate LUC from TAMs or CAMLs or Macrophages or CTCs?

7. Please define how clinicians can incorporate LUC levels in decision-making of treatment. Please elaborate more on this in discussion section.

8. Without comparison with existing biomarkers, it is unclear if LUC has any distinguishing advantages from the markers that are already existent.

Author Response

We would like to sincerely thank the reviewer for their thoughtful and constructive feedback. The insightful comments have significantly contributed to improving the clarity, depth, and robustness of our manuscript. We appreciate the reviewer’s recognition of the novelty and importance of our work in addressing predictive and prognostic biomarkers in metastatic HR+ HER2-negative breast cancer. We have carefully addressed each point raised and made substantial revisions where necessary. These changes, we believe, have strengthened the biological insights and presentation of our study while further highlighting the relevance of large unstained cells (LUC) as a potential biomarker. We are confident that the improvements will enhance the overall quality of the manuscript and are grateful for the opportunity to refine our work. Below, we provide a detailed point-by-point response to the comments.

Reviewer-2

This study addresses an important clinical challenge: the identification of predictive and prognostic biomarkers in metastatic HR+ HER2-negative breast cancer. The investigation of large unstained cells (LUC) as a novel biomarker is an innovative concept, however the lack of information on origination of LUC, validation, characterization effects the robustness of the study. LUC is the center of focus of the study, and there are no example images of appearance of LUC and how are different from other blood cells.

Comments:

  • The novelty of this study is that it explores previously underexplored biomarker LUC, however, there are lack of biological insights about the same. For example, the introduction lacks information on how LUCs are identified. Is it based on their size? LUC needs to be defined better in introduction.

Authors’ Response: We thank the reviewer for this valuable comment. In response, we have enhanced the definition and characterization of large unstained cells (LUC) in the introduction to improve clarity and biological context. We have now specified that LUCs are identified via automated blood analyzers (Siemens ADVIA). They are characterized as large, peroxidase-negative cells that remain unclassified due to their unique morphological and staining properties. These cells include precursor and atypical lymphocytes, plasma cells, or hematopoietic stem cells, as cited in previous studies (references 13-14). Their distinction from other peripheral blood cells lies in their larger size and lack of peroxidase activity.

Additionally, we emphasize the potential role of LUC in tumor immunity, linking their possible origins as precursors of CD8+ T lymphocytes to their prognostic significance. We believe this expanded information provides a clearer understanding of LUC and supports the study's rationale.

  • Additionally, document all the LUC related studies and respective cancers in a tabular form, add all the outcomes seen in patients in literature in correlation to LUC. Introduction needs significant improvement.

Authors’ Response: We sincerely appreciate this suggestion, which has allowed us to strengthen the background and provide a clearer context for the study. To address this, we have created Supplementary Table 2, summarizing all relevant LUC-related studies across various cancers and disease states, along with reported outcomes. Additionally, the introduction section has been significantly revised to better define LUC, highlight its biological characteristics, and incorporate findings from previous studies. This provides a more comprehensive rationale for exploring LUC as a biomarker in metastatic HR+ HER2-negative breast cancer.

  • In literature, has LUC been used for any prognostic value? Please re-write your introduction.

Authors’ Response: We appreciate the reviewer’s comment and agree that the prognostic value of LUC in the literature should be addressed in the introduction. In response, we have comprehensively revised the introduction to emphasize prior studies where LUC has demonstrated prognostic significance. We have included references to LUC’s role in various cancers, such as acute leukemia, chronic lymphocytic leukemia, plasma cell leukemia, and melanoma, where elevated LUC levels have been associated with adverse or favorable outcomes. These revisions highlight that LUC is a biologically relevant marker with prognostic implications across various diseases.

  • In the introduction, when you say HR+, do you mean both ER+ and PR+? Please clarify. Why was HR+ HER2- group selected specifically?

Authors’ Response: We appreciate the reviewer’s observation and have clarified the definition of HR+ in the revised introduction. Specifically, HR+ refers to breast cancers that express estrogen receptor (ER) and/or progesterone receptor (PR), as assessed through immunohistochemical analysis, in line with current ASCO/CAP guidelines. Regarding the selection of the HR+ HER2-negative group, we focused on this specific subgroup because it constitutes the largest molecular subtype of breast cancer and represents a significant clinical challenge in metastatic settings. Furthermore, CDK4/6 inhibitors combined with endocrine therapy are the current standard of care for this patient population, yet predictive and prognostic biomarkers remain limited. We hypothesized that LUC may reflect hematopoietic stem cell activity and immune status, which could influence both treatment outcomes and the incidence of adverse effects, particularly hematologic toxicities. By addressing this gap in the literature, our study aims to demonstrate the potential of LUC as a cost-effective and readily available biomarker in HR+ HER2-negative breast cancer treated with CDK4/6 inhibitors.

Patient selection: What stage were these patients? Did they receive neoadjuvant or adjuvant treatment? Did they undergo surgery? Please make this section more detailed. It lacks a lot of important information.

Authors’ Response: We thank the reviewer for highlighting the need for additional details in the patient selection section. All patients included in our study had stage IV (metastatic) HR+ HER2-negative breast cancer. At this stage, surgery does not have a role in disease control or treatment outcomes. We also clarify that prior surgeries, whether curative or palliative, were not considered to have an impact on the effectiveness of CDK4/6 inhibitors and endocrine therapy, which were the focus of this study. While some patients may have received neoadjuvant or adjuvant treatments before their disease became metastatic, these therapies were not directly relevant to the current analysis and were not included as variables influencing outcomes. We have updated the Patients and Methods section to include this clarification and believe it now provides a more comprehensive description of patient selection criteria and prior treatments.

Definition of LUC and sampling time: Same comments as introduction.

-  Explain in detail how are LUCs sampled. How do you get the cutoff 0.11 number? How is it meaningful?

- Are LUCs CAMLs?

-Are there any specific biomarkers/antibodies that are used to identify LUCs?

- What is the biological insight/function of LUCs in cancer or treatment response?

-What is the size range of LUCs? Is there a possibility of any false negatives or false positives?

- What are the different techniques for identifying LUCs? How accurate are they? Which technique was used here and why?

- How do you ensure what you are counting is LUC?

- Is it reproducible in other lab settings to identify LUC? Please give details on your protocols and methodology to help understand better how LUCs are defined.

- No images of LUCs added in paper. Add several images of LUC, validate with biomarker testing if this study is looking at LUCs indeed. How do you differentiate LUC from TAMs or CAMLs or Macrophages or CTCs?

Authors’ Response: We sincerely thank the reviewer for these insightful questions, which have allowed us to clarify the methodology, biological significance, and reproducibility of large unstained cells (LUC). Below, we address each point in detail.

How LUCs Are Sampled and the Cutoff Value:  LUCs were measured using an automated hematology analyzer (Siemens ADVIA 2120i), which identifies LUCs through a combination of peroxidase negativity and two-angle laser scatter analysis. Blood samples were collected from patients into vacuum-sealed EDTA tubes specifically designed for complete blood count (CBC) analysis. These tubes were immediately inverted several times to ensure proper mixing with the anticoagulant and then processed within two hours of collection to maintain sample integrity and accuracy of the results.

The Siemens ADVIA 2120i analyzer classified LUCs as large, peroxidase-negative cells that remained unclassified within the standard leukocyte populations. The analyzer employed:

  • Peroxidase Method: Cells were stained with a peroxidase reagent to differentiate between peroxidase-positive (e.g., granulocytes, monocytes) and peroxidase-negative cells (e.g., lymphocytes, LUCs).
  • Basophil/Lobularity Method: After red blood cell lysis, two-angle laser scatter analysis was used to evaluate cell size and complexity, allowing precise identification of large, unclassified cells as LUCs.

LUC levels were expressed as a percentage of the total white blood cell (WBC) count, as reported by the analyzer. The cutoff value of 0.11% was determined as the median LUC value in our study cohort. Based on this value, patients were stratified into two groups:

  • Low LUC group: Patients with LUC levels <0.11%.
  • High LUC group: Patients with LUC levels ≥0.11%.

This stratification enabled meaningful comparisons of clinical outcomes, including progression-free survival (PFS) and overall survival (OS), as well as hematologic toxicities, between the two groups. Routine quality control procedures and regular calibration of the hematology analyzer were performed to ensure the accuracy and reproducibility of results.

Are LUCs CAMLs? Circulating atypical mononuclear cells (CAMLs) are distinct from LUCs. CAMLs are characterized by the expression of CD45 and cytokeratins (CK8, CK18, CK19) and are typically much larger in size (21–300 μm). In contrast, LUCs are identified solely through automated hematology analyzers as peroxidase-negative cells, and they lack specific phenotypic markers, including cytokeratin expression. While CAMLs are often associated with cancer-specific immune responses, LUCs are broader in their composition, potentially including activated lymphocytes, plasma cells, or hematopoietic stem cells.

Specific Biomarkers/Antibodies for LUCs: Currently, no specific biomarkers or antibodies have been validated to identify LUCs. Their detection relies entirely on automated analyzers, which categorize them based on cell size, light scatter properties, and peroxidase activity. While biomarkers for LUCs remain an unmet need, ongoing studies using flow cytometry aim to characterize these cells further and elucidate their role in cancer biology.

Biological Insight/Function of LUCs: LUCs likely represent a heterogeneous population of cells, including precursors of T lymphocytes, plasma cells, and hematopoietic stem cells. Elevated LUC levels may reflect an active immune response or hematopoietic activity in response to systemic inflammation or malignancy. In cancer, these cells may correlate with the tumor microenvironment's inflammatory state or the body’s compensatory hematopoietic response to cytotoxic therapies.

Size Range and Potential for False Negatives/Positives: While LUCs are identified as large, unclassified cells, their exact size range is not explicitly defined by the hematology analyzer. The potential for false positives or negatives exists due to their heterogeneous nature and overlap with other cell types. However, the reproducibility of LUC measurements across clinical settings supports their reliability as a biomarker.

Techniques for Identifying LUCs: LUCs are identified through automated hematology analyzers, which provide a highly reproducible and cost-efficient approach for clinical applications. While advanced techniques like flow cytometry and immunophenotyping could offer more comprehensive phenotypic characterization, their routine use in clinical practice is limited by their complexity, resource requirements, and higher costs. Automated analyzers remain the practical standard for LUC detection, ensuring widespread accessibility and reliability in various healthcare settings.

Ensuring LUC Identification and Reproducibility: LUCs are directly quantified by hematology analyzers, ensuring reproducibility across laboratories equipped with similar devices. In this study, strict adherence to routine calibration and quality control protocols was maintained to ensure accuracy and reliability of the measurements. The hematology analyzer underwent regular internal standardization checks using control samples provided by the manufacturer, as well as external quality assurance testing through proficiency programs. These rigorous quality control measures validated the analyzer's performance and ensured consistent and accurate detection of LUC levels across all samples.

Adding Images and Differentiating LUCs from Other Cells: To address the reviewer’s request, we have included representative graphs and visuals in the supplementary material to illustrate the identification and characteristics of LUCs.

These include:

  • Original graphics obtained directly from the Siemens ADVIA analyzer, highlighting LUC classification through peroxidase negativity and laser scatter properties.

  • This image represents an example output from the Siemens ADVIA hematology analyzer used in our study. It illustrates the cell population distributions, including the identification of LUC, as part of the automated complete blood count analysis. We included this to demonstrate that the data presented in our study were generated using this device.
  • Literature-based images that provide additional context regarding the detection and potential biological roles of LUCs.

  1. Jerez J, Sanchez F, Flores F, Guajardo L, Briones JL, Selman C. Early Detection and Diagnostic Approach Through Automated Hematological Analysis for Plasma Cell Leukemia. J Med Cases. 2024;15(1):31-36. doi:10.14740/jmc4188
  2. Kutter, D., Verstraeten, L. (2000). Screening for Leukocyte Peroxidase Deficiencies by Means of Flow Cytometry: Application to the Study of Prevalence, Pathology and Genetics. In: Petrides, P.E., Nauseef, W.M. (eds) The Peroxidase Multigene Family of Enzymes. Springer, Berlin, Heidelberg. https://doi.org/10.1007/978-3-642-58314-8_19
  3. Eren, Funda Et Al. "EVALUATION OF LARGE UNSTAINED CELLS (LUC) AND NITRIC OXIDE IN DIABETES MELLITUS." Ankara Medical Journal , vol.22, no.4, pp.533-541, 2022

We acknowledge that, due to the design and functionality of automated hematology analyzers, it is not feasible to provide solitary cell images similar to those obtained via peripheral blood smear microscopy. The Siemens ADVIA analyzer differentiates LUCs from other cells based on peroxidase negativity and basophil/lobularity parameters, focusing on size and light scatter characteristics rather than individual cell imaging or marker-based identification.

Regarding differentiation from other cell types (e.g., tumor-associated macrophages, circulating atypical mononuclear cells, circulating tumor cells), the analyzer does not provide specific phenotypic information since it does not utilize antibody-based staining or marker-level analysis. Advanced methods such as flow cytometry or additional staining techniques are required for precise characterization of LUCs at a molecular level. However, these methods were beyond the primary aim of our study, which focused on evaluating the clinical significance of LUC levels in relation to treatment outcomes and hematologic toxicities.

Please define how clinicians can incorporate LUC levels in decision-making of treatment. Please elaborate more on this in discussion section.

Authors’ Response: We thank the reviewer for this important comment, which allows us to better contextualize the clinical utility of LUC levels. We have elaborated on this point in the discussion section as follows: LUC levels may serve as a practical and easily accessible biomarker to guide clinical decision-making in patients with metastatic HR+ HER2-negative breast cancer receiving CDK4/6 inhibitors. Our study demonstrates that patients with higher LUC levels exhibit a lower incidence of hematologic toxicity and experience prolonged progression-free survival (PFS) and overall survival (OS).

Clinically, this observation has two key implications

  • Toxicity Monitoring and Management: Patients with lower LUC levels may be at a higher risk for hematologic adverse events, such as grade ≥3 neutropenia. For these patients, clinicians may consider closer monitoring with more frequent complete blood count (CBC) assessments and earlier interventions, such as dose reductions, prophylactic measures, or supportive therapies like granulocyte-colony stimulating factor (G-CSF).
  • Treatment Follow-Up and Prognosis: Elevated LUC levels could be used as an indicator of favorable prognosis, potentially informing clinicians about expected treatment outcomes, including longer PFS and OS. Patients with high LUC levels may require less intensive toxicity monitoring, allowing resources to be allocated more efficiently.

Given the ease of measuring LUC levels through automated CBC analyzers, this biomarker is readily reproducible and could be incorporated into routine clinical practice without additional costs. However, prospective validation studies are needed to confirm its role as a predictive and prognostic marker and to establish standardized thresholds for clinical use.

We believe this addition clarifies the translational potential of our findings and provides clinicians with a rationale for integrating LUC levels into treatment monitoring and decision-making.

Without comparison with existing biomarkers, it is unclear if LUC has any distinguishing advantages from the markers that are already existent.

We thank the reviewer for this important observation. While LUC has not been directly compared to established pathological biomarkers such as HER2 status in predicting CDK4/6 inhibitor efficacy, it offers certain unique advantages that distinguish it as a potential clinical tool.

  • Accessibility and Cost-Effectiveness: LUC levels are obtained directly through routine automated CBC analyses, making it a low-cost, widely accessible biomarker. This is in contrast to many existing pathological or molecular biomarkers that often require specialized assays, such as immunohistochemistry or genomic sequencing, which may not be readily available in all clinical settings.

  • Unique Role in Hematologic Toxicity Prediction: Unlike HER2 status or other known markers, LUC is the only marker currently reported to predict hematologic toxicities, specifically grade ≥3 neutropenia, associated with CDK4/6 inhibitors. This dual prognostic and predictive role for both survival outcomes and toxicity monitoring makes LUC particularly valuable in optimizing treatment decisions.

While LUC represents a promising and practical biomarker, we acknowledge that further prospective studies are required to validate its superiority or complementary role alongside existing biomarkers. We have incorporated these clarifications into the discussion section to better highlight LUC's advantages and limitations.

We would like to sincerely thank Reviewer 2 for their insightful and detailed feedback. Your constructive comments have helped us improve the biological context, methodology, and overall clarity of the manuscript. We hope that our revisions have sufficiently addressed your suggestions and strengthened the scientific merit of our work.

Reviewer 3 Report

Comments and Suggestions for Authors

This manuscript presents an interesting and novel investigation into the prognostic and predictive role of large unstained cells in metastatic hormone receptor-positive HER2-negative breast cancer treated with CDK4/6 inhibitors. The study showed that high LUC levels are associated with longer PFS and OS, as well as lower incidence of grade ≥3 neutropenia, indicating that LUC may be a promising biomarker for outcomes and toxicity management in this patient population.

Limitations and Suggestions:

Alternative Treatment Options: Alternative treatment options (i.e. surgery) may influence outcomes. Please cite PMID: 36551722 in your Discussion section to improve the quality of your manuscript.

LUC Characterization: Lack of flow cytometric analysis for precise LUC characterization limits biological interpretation. Incorporation of this analysis in further research would enhance mechanistic understanding.

Heterogeneity in CDK4/6 Inhibitors: Differences in hematologic toxicity among inhibitors are recognized but not deeply explored. Stratified analysis by inhibitor type may provide further insights.

Limited Follow-Up Duration: The median follow-up of 18.5 months is relatively short for robust OS conclusions in this setting. Longer-term data would strengthen the findings.

Author Response

We sincerely thank the reviewer for their thoughtful and constructive feedback on our manuscript. We greatly appreciate the positive recognition of our study's novelty and its contributions to understanding the prognostic and predictive role of large unstained cells (LUC) in metastatic HR+ HER2-negative breast cancer. The insightful suggestions provided have highlighted key areas for refinement and future exploration, and we believe they have further strengthened the scientific quality and clarity of our work. We have carefully addressed each comment and limitation raised, and we hope that the revisions and clarifications provided below adequately reflect our gratitude for the reviewer’s valuable input.

This manuscript presents an interesting and novel investigation into the prognostic and predictive role of large unstained cells in metastatic hormone receptor-positive HER2-negative breast cancer treated with CDK4/6 inhibitors. The study showed that high LUC levels are associated with longer PFS and OS, as well as lower incidence of grade ≥3 neutropenia, indicating that LUC may be a promising biomarker for outcomes and toxicity management in this patient population.

Limitations and Suggestions:

Alternative Treatment Options: Alternative treatment options (i.e. surgery) may influence outcomes. Please cite PMID: 36551722 in your Discussion section to improve the quality of your manuscript.

Authors’ Response: We thank the reviewer for this valuable suggestion. We have now incorporated the cited study (PMID: 36551722) into the manuscript to acknowledge the role of alternative treatment options, such as surgery, in influencing outcomes. In metastatic HR+ HER2-negative breast cancer, the role of surgery is limited and does not typically influence long-term outcomes when systemic therapy is the primary treatment strategy. However, we recognize that locoregional treatments, including surgery, may have potential benefits in select patients and have referenced this important study to ensure a more balanced and comprehensive manuscript.

We believe this addition enhances the quality of the manuscript by addressing alternative treatment options and their potential impact on clinical outcomes.

LUC Characterization: Lack of flow cytometric analysis for precise LUC characterization limits biological interpretation. Incorporation of this analysis in further research would enhance mechanistic understanding.

Authors’ Response: We thank the reviewer for this insightful comment, which highlights an important area for future research. While flow cytometric analysis would indeed provide a more precise characterization of LUCs, it is noteworthy that, despite the extensive literature on LUCs, such detailed searches have not yet been widely performed. In our current study, LUC levels were directly measured using automated hematology analyzers, which offer practical, reproducible, and cost-effective detection for clinical purposes.

We fully agree that incorporating flow cytometric assessment of LUCs would enhance the mechanistic understanding of their biological significance in cancer and treatment response. As such, we plan to address this limitation in our future research, where we will specifically focus on flow cytometric characterization of LUCs to better define their immunophenotypic profile and clarify their role in tumor biology and hematologic toxicities.

We have added this point to the Discussion section to acknowledge this limitation and outline our plans for future studies.

Heterogeneity in CDK4/6 Inhibitors: Differences in hematologic toxicity among inhibitors are recognized but not deeply explored. Stratified analysis by inhibitor type may provide further insights.

Authors’ Response: We thank the reviewer for highlighting this important point. As presented in Table 5, our multivariate analysis demonstrated that, independent of LUC levels, the use of palbociclib and fulvestrant was associated with a significantly higher risk of grade ≥3 neutropenia. This finding aligns with existing real-world evidence, which similarly reports higher hematologic toxicity rates with palbociclib compared to other CDK4/6 inhibitors.1

While a stratified analysis by inhibitor type was not the primary aim of this study, we agree that exploring the differences in toxicity profiles among CDK4/6 inhibitors could provide further insights into their clinical use. We have acknowledged this in the Discussion section and emphasized the need for larger, stratified analyses in future studies to better delineate the relationship between LUC levels, inhibitor type, and toxicity risk.

Limited Follow-Up Duration: The median follow-up of 18.5 months is relatively short for robust OS conclusions in this setting. Longer-term data would strengthen the findings.

Authors’ Response: We appreciate the reviewer’s observation and agree that a longer follow-up duration would further strengthen the robustness of the overall survival (OS) conclusions. However, it is important to note that the absence of disease progression within the first 12 months of CDK4/6 inhibitor treatment has been associated with significantly longer OS and subsequent progression-free survival, as demonstrated in prior studies.2

In our cohort, the observed trends in OS and PFS align with these findings, suggesting that early responses to CDK4/6 inhibitors are predictive of long-term outcomes. We have highlighted this point in the Discussion section to provide additional context and have emphasized the need for extended follow-up in future studies to validate and expand upon our findings.

References

  1. Lin, W., Zeng, Y., Weng, L.et al. Comparative analysis of adverse events associated with CDK4/6 inhibitors based on FDA’s adverse event reporting system: a case control pharmacovigilance study. BMC Pharmacol Toxicol 25, 47 (2024). https://doi.org/10.1186/s40360-024-00770-6
  2. Munzone E, Pagan E, Bagnardi V, Montagna E, Cancello G, Dellapasqua S, Iorfida M, Mazza M, Colleoni M. Systematic review and meta-analysis of post-progression outcomes in ER+/HER2- metastatic breast cancer after CDK4/6 inhibitors within randomized clinical trials. ESMO Open. 2021 Dec;6(6):100332. doi: 10.1016/j.esmoop.2021.100332. Epub 2021 Dec 1. PMID: 34864350; PMCID: PMC8645913.

We extend our sincere thanks to Reviewer 3 for the valuable comments and thoughtful observations. Your feedback allowed us to better contextualize our findings and address key areas for improvement, particularly in the discussion and methodology sections. We trust that our revisions have clarified your concerns and enhanced the quality of the manuscript.

Round 2

Reviewer 2 Report

Comments and Suggestions for Authors

Thank you for addressing my comments 

Reviewer 3 Report

Comments and Suggestions for Authors

The manuscript can be accepted in the present form